# Understanding the Functionality and Burden on Decentralised Rural Water Supply: Influence of Millennium Development Goal 7c Coverage Targets

**Jonathan P. Truslove [1],\* , Alexandra V. M. Miller [1], Nicholas Mannix [1], Muthi Nhlema [2], Michael O. Rivett [1], Andrea B. Coulson [3], Prince Mleta [4] and Robert M. Kalin [1]**

[1] Department of Civil and Environmental Engineering, University of Strathclyde, Glasgow G1 1XJ, UK; alexandra.miller@strath.ac.uk (A.V.M.M.); nicholas.mannix@strath.ac.uk (N.M.); michael.rivett@strath.ac.uk (M.O.R.); robert.kalin@strath.ac.uk (R.M.K.)

[2] BASEflow, Galaxy House, Blantyre, Malawi; muthi@baseflowmw.com

[3] Department of Accounting and Finance, University of Strathclyde, Glasgow G4 0QU, UK; a.b.coulson@strath.ac.uk

[4] Ministry of Agriculture, Irrigation and Water Development, Government of Malawi, Tikwere House, Lilongwe, Malawi; princemleta@gmail.com

\* Correspondence: jonathan.truslove@strath.ac.uk; Tel.: +44-(0)141-548-3275

**Abstract:** The sustainability of rural groundwater supply infrastructure, primarily boreholes fitted with hand pumps, remains a challenge. This study evaluates whether coverage targets set out within the Millennium Development Goals (MDG) inadvertently increased the challenge to sustainably manage water supply infrastructure. Furthermore, the drive towards decentralised service delivery contributes to the financial burden of water supply assets. A sample size of 14,943 Afridev hand pump boreholes was extracted from a comprehensive live data set of 68,984 water points across Malawi to investigate the sustainability burden as emphasis shifts to the 2030 agenda. The results demonstrate that the push for coverage within the MDG era has impacted the sustainability of assets. A lack of proactive approaches towards major repairs and sub-standard borehole construction alongside aging infrastructure contributes to reduced functionality of decentralised supplies. Furthermore, costly rehabilitation is required to bring assets to operational standards, in which external support is commonly relied upon. Acceleration towards the coverage targets has contributed towards unsustainable infrastructure that has further implications moving forward. These findings support the need for Sustainable Development Goals (SDG) investment planning to move from a focus on coverage targets to a focus on quality infrastructure and proactive monitoring approaches to reduce the future burden placed on communities.

**Keywords:** borehole; decentralisation; functionality; hand pump; low-income regions; maintenance; Malawi; rehabilitation; sustainability

---

## 1. Introduction

In 2015, the Millennium Development Goals (MDG) ended. Between 2000 and 2015, the MDG target for water and sanitation (7c) aimed "to halve the proportion of people without sustainable access to safe drinking water and basic sanitation". According to the World Health Organisation (WHO)/United Nations Children's Fund (UNICEF) Joint Monitoring Programme (JMP) for Water Supply and Sanitation, this target was globally delivered in 2010. Target achievement was highly influenced by large, populous countries such as China and India, with Sub-Saharan Africa (SSA) lagging behind [1]. By 2015, 91% of the global population was reported to be using an improved

drinking water supply, such as piped water, boreholes, protected wells and springs or rainwater. However, 663 million people did not have access to an improved supply, with half of these people living in SSA [2]. In 2015, all member states of the United Nations General Assembly agreed to "the 2030 Agenda", which aims to "end poverty in all forms", to "shift the world on to a sustainable and resilient path" and to ensure "no one will be left behind". Within the 17 Sustainable Development Goals (SDG) that were established, SDG 6 aims to "ensure availability and sustainable management of water and sanitation for all" and displays an increased focus on global water and sanitation challenges. The commitment to "leave no one behind" will require an increased focus on sustainable supplies for disadvantaged groups in low-income regions drawing on lessons learned from the coverage approach of the MDGs.

The JMP indicators have since been updated to reflect the SDGs, and to include additional criteria for water supply service levels. The 2017 progress report on the SDG highlights how the SDGs address the unfinished business from the MDGs and raise the bar with new monitoring categories [3]. At the high political forum in July 2018, the progress of SDG 6 was under review, highlighting the proportion of the population that meet these new monitoring criteria and the challenges of meeting a sustainable water future [4]. This review further highlights the continued slow progress of SSA towards global targets as 38% of the population do not even have a basic level of service, compared to 12% of the overall global population in 2015.

While the JMP confirms positive progress in rural water supply usage during the MDGs, concerns have been raised that these indicators actually hide a low level of service, as further discussed by Adank et al. [5]. In rural SSA, an estimated 184 million people rely on hand pumps for domestic supply [6], with approximately two out of three hand pumps working at a given time [7]. Across SSA, the Afridev hand pump is the most widely recognised village-level operations-and-maintenance (O&M) pump [6]. While the installation of such improved supplies has improved the coverage of the region, the reliability of hand pump supplies remains a concern. The challenges of maintaining hand pumps at the rural level, including the Afridev, are well established and obstructed by more than just maintenance issues [8]. When an improved source is unreliable, unaffordable or contaminated, and potentially leading to abandonment, the national monitoring statistics may not reflect this reality at the local level [9–11]. Affected water users may depend on unimproved supplies or surface-water sources, temporarily or definitively. This will ultimately have a considerable effect on the health and overall poverty reduction of the region, that has compromised SSAs efforts in relation to the MDGs and will continue to hinder progress towards the SDGs.

This paper argues that the international push to meet the coverage targets of MDG target 7c resulted in the "sustainable access" aspect of this target falling short. It is contended that poor standards of water supply infrastructure installed to increase coverage during the MDG period have left rural populations in low-income regions with—or vulnerable to—the burden of maintaining the supply at the local level, contradicting national policies and hindering progress towards the SDGs. If SDG 6 is to be robustly met, appropriate targets must be set to ensure sustainable access. This includes essential monitoring of assets across the country [4], in which comprehensive management information systems (MIS) are useful tools [12,13]. This is essential for sustaining aging infrastructure that has been subject to a major coverage goal initiative. It is vital to look beyond simple indicators of coverage and consider the quality of the service provision [5] to further understand and enable proactive approaches that will sustainably manage assets [14–17].

This paper investigates the Afridev hand pump boreholes that have been installed in 25 out of the 28 districts of Malawi during the MDGs, drawn from a large and recent national data set. Here we investigate the functionality of Afridev hand pump boreholes installed in Malawi that have been subject to the major coverage-driven initiative of the MDGs, and to review if the reliance on reactive approaches to maintenance are a sustainable solution for decentralised service providers. We found that the acceleration to meet the coverage targets of the MDGs contributed to unsustainable infrastructure, alongside the challenge of maintaining aging infrastructure that has contributed to the burden of

sustaining water supply assets in the rural communities of Malawi. This provides evidence for water policy updates with associated guidance to practitioners on the impacts to long-term sustainability.

## 2. Methodology

### 2.1. Study Area

Malawi is a land locked country in SSA located between 9° S and 17° S (latitude) and 33° E and 36° E (longitude), bordered by Mozambique in the east, south and west, Tanzania in the northeast and Zambia in the northwest. Malawi has a population of 18.6 million, with approximately 84% located in a rural setting and with more than 50% of the population living below the poverty line. Malawi's population is set to double by 2030, placing greater pressure on the country's water sector. Climate impacts further risk the provision of safe water as Malawi is prone to droughts and floods, with the main rainy season occurring between November to April and two dry seasons during the rest of the year. Furthermore, there is less than 1400 m$^3$/year/person of available total water resources, making Malawi one of the most water-scarce countries in the world [18].

In rural settings there is a reliance on groundwater for the main source of daily water needs, and for social and economic development. Boreholes fitted with Afridev hand pumps are the main technology used for rural water supply. The Afridev hand pump emerged as the dominant hand pump in Malawi through standardisation in the 1990s [6], and is an approved and recognised technology by the Government of Malawi. Rural communities manage their water supplies under the community-based management (CBM) approach, as the Government of Malawi's National Water Policy promotes the decentralisation of rural water supplies to local governments, and further reflects the coverage targets set out in the MDGs. However, due to the limited capacity of local governments to develop the rural water sector, there is an over reliance on external funding and support [19].

### 2.2. Data Collection

The Scottish Government Climate Justice Fund (CJF) Water Futures Programme has been working in partnership with the Government of Malawi evaluating the sustainability of rural water supplies since 2011. The programme aims to support the Government of Malawi to achieve SDG 6. A core workstream of the programme is to evaluate the sustainability of all rural water supply assets across Malawi. Information regarding the viability and sustainability of each asset is collected through a water point functionality survey based on SDG 6 indicators and the additional needs of the Government of Malawi. These include, but are not limited to, geographical information on water points and subsequent communities served, installation dates, donors and installers, service provider asset and water point O&M, accessibility, reliability and functionality status of the water point (www. cjfwaterfuturesprogramme.com).

This study draws on the data collected by the CJF and collated into the bespoke developed MIS, mWater (www.mwater.co). Of the various MIS available, mWater was chosen as the preferred asset analysis platform due to a wide range of capacity and adaptability [11,20]. Rural water asset status data from 2011–2016 was collected under subcontract by the Non-Government Organisation (NGO) Water for People using AkvoFLOW (akvo.org/products) across 8 Southern Region districts of Malawi. A further mapping exercise was conducted across these Southern Region districts of Malawi in 2017–2018 using mWater. Since 2018, the programme has embarked on evaluating the asset status of every water point across Malawi, expecting to complete in 2019. All available historic data were subsequently imported into the mWater database.

The Government of Malawi staff go through classroom and field training to teach them how to respond to surveys which are provided in both English and Chichewa (local language). During this training they assess the functionality of the Afridev hand pump based on technical specifications. The additional data collected, such as the flow rate, assist with the functionality definition alongside design specifications.

Though the Government of Malawi staff are trained on data collection and the subsequent data undergoes rigorous quality assurance checks, non-sampling errors cannot be excluded. However, the field data that is uploaded to the server undergoes two rounds of approvals before being accepted. These include extensive quality checks of the uploaded data, such as the accuracy of the mapped locations, and rejects any submissions that do not meet these quality checks. The rejection rate during approvals is less than 5% of the submitted data, and the monitoring of the accepted data is subject to thorough quality checks. Historic data imported into the data set has undergone any site deduplication, increasing the accuracy and reliability of long-term trends in the database.

*2.3. Sampling and Methods*

This study investigates the functionality status of water points installed in 25 out of the 28 districts in Malawi across the MDG period 2000–2016. It is recognised that when considering mapping and monitoring data, the term "functionality" is a snapshot temporal indicator for sustainability, which is a multi-variable service over time, discussed at length by Carter and Ross [21]. The term partial functionality has been introduced over recent years due to the need to further define functionality [21,22]. This concept incorporates many different situations that affect the overall performance of the water supply including maintenance issues, water quality and variations in yield. Additional supporting information is required to further support the classification of the status of water point assets, thus a wider range of information is included in the CJF Water Point Functionality Survey data set providing insights into the current burden of the MDG infrastructure in Malawi. Furthermore, it highlights the importance of continued and improved monitoring of water points and influencing factors to ensure the long-term sustainability of the Government of Malawi's assets.

Therefore, for avoidance of doubt, the term "functional" is used here to describe a water point that is in operational condition and providing water according to design specifications, "partial functionality" is a water point that is providing water, but in a reduced capacity (e.g., only certain times of the year, not according to flow rate specifications, changes in site conditions, repairs required, etc.), and "non-functional" is used to describe a water point that no longer provides water on a regular basis at the time of the asset audit. These definitions, alongside the additional information collected, allow for problem areas or areas of need to be highlighted for assistance with decision making. The definitions are adopted across the MIS mWater (see Section 2.2) that is used by NGOs and others.

From the 68,984 water points mapped by the CJF to date (January 2019), a subset of 23,073 drilled boreholes equipped with Afridev hand pumps were captured from the mWater live database. The data was filtered to those points installed during the MDG period between 2000 and 2015, and 2016 for the transition to the SDGs. This resulted in a data set of 14,943 Afridev hand pump boreholes. The distribution of this data set is shown in Figure 1 with further detailed information provided in Table A1 (see Appendix A).

From this data set, further investigation was made into (a) whether or not the supply had a service provider present for O&M, and (b) if available, the service provider was decentralised (area or water mechanic, community members, an institution, local government, NGO, self-supplied, public operator, water point committee or water user association). Where a response was recorded as "don't know" for a present service provider, the data was omitted from that part of the analysis to ensure only supplies with decentralised service providers were considered.

Capital maintenance expenditure (CapManEx) is described as going beyond routine O&M from a service provider to repair and replace assets that keep them running [23], which risks becoming an issue if not addressed through an asset's life cycle [24]. This accounts for major repairs which are crucial for the sustainability of a service and rehabilitation. Investments into rehabilitation treat the costs as the start of a new service, as they are required to bring the existing systems back to operational use if appropriate O&M is not conducted [25], but rehabilitation is seldom considered or practiced by communities or local government due to the costs involved. This further variable was evaluated across

the data set to investigate (a) when rehabilitation was conducted by date (2000–2018), (b) the status of functionality from the result of the rehabilitation exercises and (c) who funded these rehabilitations. Within this data set, rehabilitation was defined as a single major repair consisting of 1,500,000 MK (Approximately 2062 USD, where 1 USD = 727 MK, as of January 2019). This monetary definition was defined by the Government of Malawi at the beginning of the 2017 mapping exercise. This variable was also constrained to the Afridev hand pump boreholes installed during the 2000–2016 timescale.

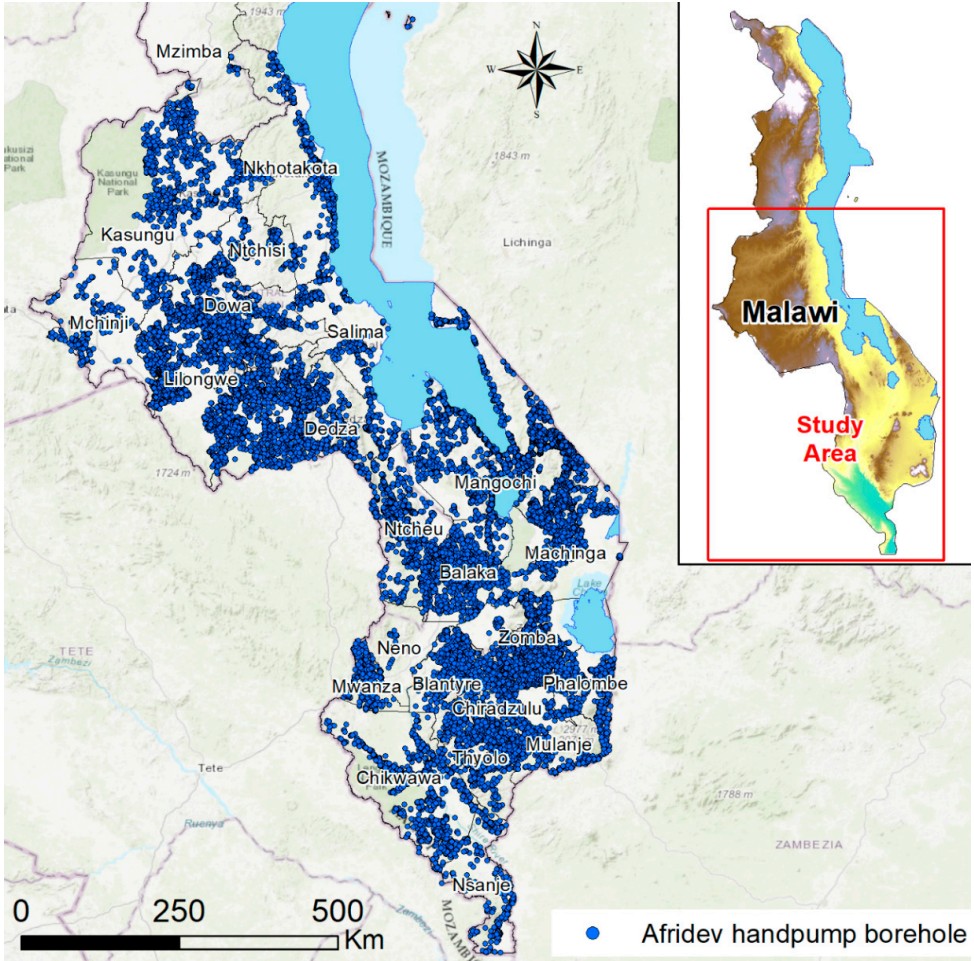

**Figure 1.** Afridev hand pump boreholes installed between 2000–2016 mapped to date by the CJF (n = 14,943 as of January 2019, with mapping generally proceeding from south to north).

## 3. Results and Discussion

### 3.1. Installation of Millennium Development Goal Afridev Hand Pump Boreholes

The MDG agenda saw an increase in improved access to drinking water, globally meeting the target by 2010, but with SSA falling behind [1]. However, according to the JMP monitoring data, while SSA failed to meet the target, Malawi met the MDG coverage target by 2015 [2]. Approximately 84% of the population of Malawi are located in a rural setting. According to the JMP, Malawi has shown a positive shift in rural water supply coverage with an initial 61% coverage in 2000 rising to an 85% coverage of improved supplies by the end of 2015 (63% at least basic and 20% limited). The reality in Malawi is a substantial increase from 49 to 75% usage of non-piped improved supplies, which are predominantly hand-pumped groundwater supplies, with a decrease of 12 to 10% of piped improved supplies [2]. Figure 2 presents the number of Afridev hand pump boreholes installed across Malawi through the 2000–2016 period. The data set supports that that the MDG targets led to an increase in

new water supply installations across Malawi to increase coverage targets, with specific increases of coverage evident at various dates across the MDG period.

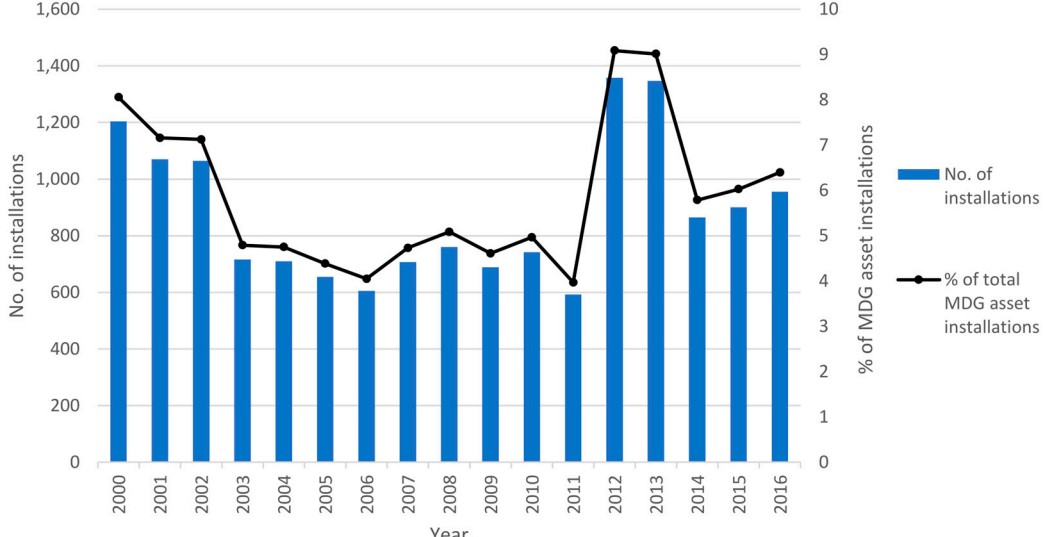

**Figure 2.** Trend of Afridev hand pump borehole installations across Millennium Development Goal (MDG) key dates in Malawi (n = 14,943).

The first key note of investment into MDG coverage expresses a large number of Afridev hand pump borehole installations at the beginning of this period (2000–2002), followed by a reduced rate up until 2011, before an increase in the years approaching to the end of the MDG period, with a marked jump in installations at 2012–2013 (see Figure 2). The MDG target for water, MDG 7c, was repeatedly edited until 2006 as further discussed by Bartram et al. [26]. However, the target during this period and after its final adoption always had an emphasis on coverage for drinking water, with sanitation being added after 2002 [27]. This increase of installations followed by a decline from 2002 onward suggests a response to a change in the coverage targets of MDG 7c, with a joint focus on drinking water and sanitation. However, JMP states that in 2010, as SSA was falling behind but the world was on track for MDG 7c, Malawi was one of the few countries within SSA that fell into the category of "on track" and progressing well towards MDG 7c by 2010 [1], with approximately 600–1200 installations per year up until that date evident in Figure 2.

The marked increase of 2012 and 2013 installations obvious in Figure 2 is attributed to the response to the introduction of Malawi's Water Resources Act 2013, the JMP progress reporting for Malawi in 2010 and the observed doubled effort compared to the 10 years prior to meet MDG 7c by 2015. As the MDGs gained momentum, there was a greater interest placed on policies and financing [26], including an increase of regional policies. In the case of Malawi, policy and guidelines were developed with an aim of reflecting the requirements of the MDGs [28,29]. However, the rapid response for coverage targets may have caused vulnerabilities and led to potential negative impacts, particularly to the long-term post-construction sustainability of the installed systems where policy primarily focused on improved access and CBM approaches.

The findings in Figure 2 are consistent with wider sector observations by the Rural Water Supply Network (RWSN). The RWSN Executive Steering Committee [7] describes the tendency for actors in the development sector to commit significant amounts of funding to new water supply infrastructure with the focus of fulfilling numerical coverage targets. This subsequent push for coverage during the MDG era to meet MDG 7c, and increased investment from NGOs and social actors, lacked the investment into capacity development to maintain supplies for disadvantaged groups—an old and leading concern [30,31]. The post-construction period is often considered "somebody else's problem" that has contributed to a burden of water supply assets across rural Malawi.

Rural water supply infrastructure can be complicated to maintain sustainably, and may often fall into disrepair before its anticipated design life expectancy, while infrastructure may be implemented regardless of considerations as to whether it can be kept operational [7]. As such problems in the rural water supply sector were gradually becoming more evident towards the end of the MDG period, this posed sustainability concerns moving forward from the year the MDG target for water was declared fulfilled in consideration of potentially many installed boreholes from the MDG era. The lack of long-term sustainability planning when implementing new assets risks having those assets fall into disrepair soon after installation before inevitably requiring the funding of large-scale rehabilitation to bring them back to operational use or risking complete failure and abandonment [25].

### 3.2. Trends in Post-Construction Service Provision and Functionality

In rural Malawi, the service delivery of water is focused on decentralisation, primarily through CBM [29]. This model is designed with the aim to benefit and empower communities [32]. Under CBM, the O&M management and financial responsibilities of water supply falls to the voluntary participation of the community. The most common approach for CBM is for capital costs to be primarily covered by external aid while 100% of O&M costs are community-owned. Research internationally has shown that the challenges service providers face to accommodate this, alongside the lack of willingness to pay for the financing of O&M within a community, have led to an array of additional problems and the approach being questioned [7,8,33–40].

Table 1 presents the statistical overview of the extracted data set. Where service providers were present, the average functionality across the MDG years (71.44%) proved to be slightly higher than the trend expressed in literature, of two-thirds of installed hand pumps working at any given time. However, when rehabilitated supplies were excluded, the functionality became 66.58%, confirming the trend. The average functionality in Table 1 was influenced by the functionality of newer systems that steadily decreased as systems age (i.e., depreciation of the systems contributed to the lower functionality of older systems).

**Table 1.** Functionality at the time of audit of Afridev hand pump boreholes installed between 2000–2016, evaluating the influence of service providers present and rehabilitation conducted during the life cycle. Percentage values in parentheses are in relation to the total n.

| Service Provider Present | Variable | n | Mean over 2000–2016 % | Min. Annual Average Functionality between 2000–2016 % | Max. Annual Average Functionality between 2000–2016 % |
|---|---|---|---|---|---|
| Yes | No rehabilitation conducted during life cycle | 12,476 | 70.77 (66.58) | 62.78 (56.41) | 84.14 (81.72) |
| | Rehabilitation conducted during life cycle | 805 | 78.43 (4.86) | 66.67 (1.34) | 91.43 (11.09) |
| | Total | 13,281 | 71.44 | 63.85 | 84.25 |
| No | No rehabilitation conducted during life cycle | 965 | 56.91 (55.03) | 36.84 (34.43) | 80.30 (80.30) |
| | Rehabilitation conducted during life cycle | 35 | 77.16 (2.58) | 0 (0) | 100 (5.75) |
| | Total | 1000 | 57.61 | 38.98 | 80.95 |

It is well established that post-construction management is essential to ensure its continuous service delivery of improved supplies. However, as the MDG era strived for drinking water coverage and Malawi's policy to promote service delivery at the CBM level [28], the sustainability and performance of these systems is questionable, as shown by the functionality distribution where service providers are present in Table 1. Further research supports this observation, as financing and conducting preventative maintenance can appear a redundant exercise to service providers [15,36,41]

when they are an essential part of a water systems life cycle, and a more cost-effective strategy than an often-repeated rehabilitation exercise [42].

Table 1 shows that over the MDG era, Afridev hand pump boreholes could be functional for extended periods without a service provider to conduct O&M (55.03%). This average consists of systems dating to the beginning of the MDGs, suggesting that when water points are well constructed, they can operate for years without issue. While this improves sustainability, the presence of a service provider can significantly improve the functionality of a system as shown by the minimum percentages expressed in Table 1. However, service providers commonly struggle to undertake major repairs to maintain aging infrastructure that will ultimately require rehabilitation. This is supported by Figure 3, in which the functionality of Afridev hand pump boreholes with service providers is presented.

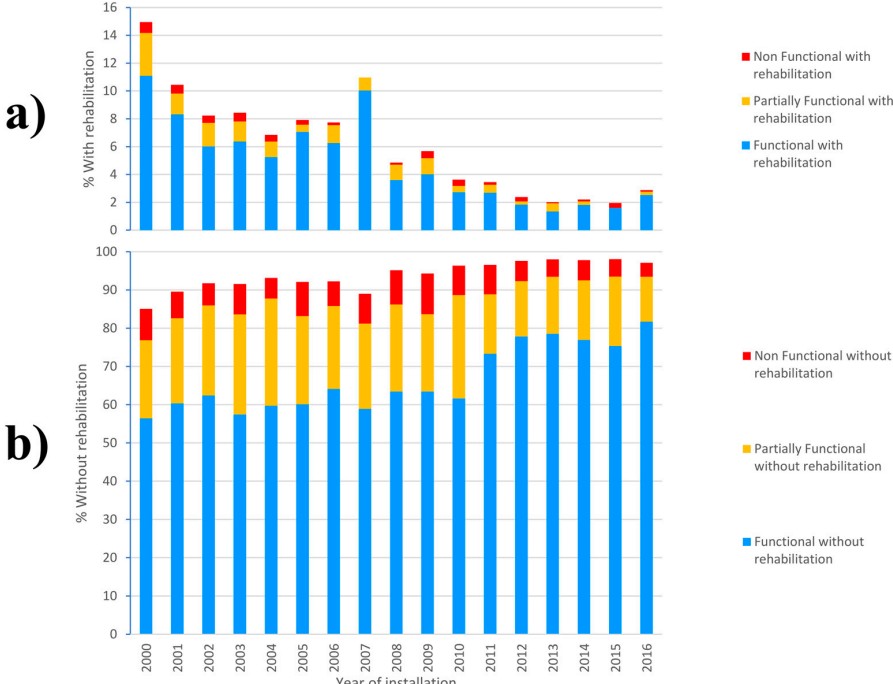

**Figure 3.** Afridev hand pump borehole functionality distribution across MDG period with service providers (n = 13,281). (**a**) % of data set with rehabilitation conducted during the life cycle; (**b**) % of data set without rehabilitation conducted during the life cycle.

The functional status confirms a higher percentage of fully functional Afridev hand pump boreholes between 2011 and 2016 (Figure 3), which is above the two-thirds trend found in the literature. This is preceded by a sharp reduction in 2010 that is less marked towards the start of the MDG period, further supporting the trend that hand pumps typically last 5 years without appropriate O&M [43]. This suggests that between 2000 and 2010, service providers were conducting the appropriate O&M required to maintain functionality, conforming to the trends expressed in literature.

The decline in functionality as the system age increased, alongside the subsequent increase in the rehabilitation of potentially partially functional or non-functional early MDG systems, highlights the lack of post-construction support for service providers who conducted routine O&M and were subjected to a depreciation of supplies requiring CapManEx. Furthermore, the focus on rapid provision of water without consideration of post-construction O&M support had an impact on the sustainability of the supplies. "One-time investment" approaches adopted by NGOs and donors, and investments into new assets, risk leaving service providers who struggle to provide the maintenance and major repairs required for sustainability unsupported. The argument that "communities are always capable of managing their own facilities on their own" has been widely criticised [7], as has the debated CBM model since implementation. This can lead to a reduction of O&M or abandonment of service

provision. A study by Hutchings et al. [34] highlights that success is possible when the model takes a more professional approach to manage the complexities of rural water supply with an emphasis on external support.

Figure 4 presents the functionality of Afridev hand pump boreholes without service providers. These results further indicate that a rapid provision of water points (see Figure 2) has an impact on long-term sustainability, as no service providers were present, especially in the cases of those installed during the late MDG era. While the late MDG era boreholes suggest a rush for coverage without the necessary service provision capacity, it should also be considered that in installations from the early MDG period (Figure 4), water points may have once had a service provider that later departed.

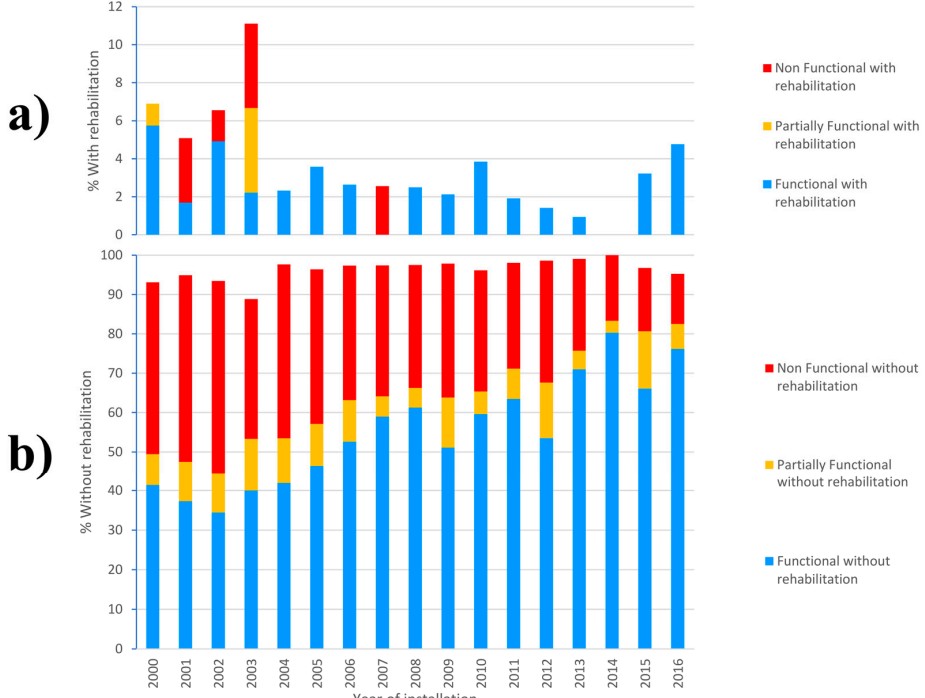

**Figure 4.** Afridev hand pump borehole functionality distribution across MDG period without service providers (n = 1000). (**a**) % of data set with rehabilitation conducted during the life cycle; (**b**) % of data set without rehabilitation conducted during the life cycle.

Generally, communities experience a reliable service after a new system is installed and a slightly reduced service after a few years as breakdowns begin to occur caused by a depreciation of infrastructure due to continued usage without appropriate O&M. Both the functionality and corresponding non-functionality expressed in Figure 4 highlight how unmaintained systems are affected by the depreciation of infrastructure. This trend supports previous interpretations by Foster [8], who has described statistically significant predictors of non-functionality, and presented strong evidence of the negative relationship between system age and functionality. This further supports the importance of establishing O&M service provisions within post-construction support. However, when water points are well constructed, they have the potential to operate for years without issue. This is supported by Figure 4, which shows the presence of functional supplies installed during the MDG period, although the minimum expressed during this period was 34.43% in 1 year. However, this significant reduction of functionality due to depreciation during the MDG period risks the further reduction of functional supplies into the SDGs without adequate service provision.

Figure 3 suggests that while routine O&M was present beyond the 5-year mark, ultimately the limited CapManEx (or major repairs) conducted on aging supplies contributed to a reduced service that risked breakdown (i.e., partial functionality). When comparing this trend to Figure 4, the impact of depreciation is more noticeable. The lack of service provision to conduct O&M or CapManEx

contributed to breakdown (non-functionality), rather than a reduced service (partial functionality). In reality, the costs of CapManEx often go unconsidered, even within service provision, which further contributes to the reduced service of the MDG assets. The non-functionality expressed in both Figures 3 and 4 risk costly rehabilitation to bring assets back to an operational standard, which is often beyond the capability of decentralised service providers to finance. These findings support the premise that sustaining functionality is heavily influenced by the institutional post-construction support and the depreciation of infrastructure.

### 3.3. Quality of MDG Infrastructure

The decline in functionality across the MDG era further supports the need for post-construction support to reduce the risk of premature failure of water supply assets. However, the partial functionality and non-functionality of boreholes installed towards the end of the MDG period point to a risk of poorly designed sub-standard installations that have contributed to a reduced service and abandonment of assets. Falkenmark [44] (p.15), states at the beginning of the International Drinking Water Supply and Sanitation Decade (1981–1990), "If well-drilling and hand pump problems are focused during the first half of decade, it is probable that the operation and maintenance problems will be the ones dominating during the second half." However, issues with sub-standard borehole construction are still relevant across the assets installed during the MDGs. While there is a focus on hand pumps and institutional capacity building when planning the sustainability of rural water supplies, the effects of sub-standard borehole construction due to accelerating the MDG coverage likely contributes to excessive delivery costs for the rural service provision.

It is proposed here that borehole construction quality is an important contributing factor towards water point functionality and sustainability, alongside hand pump O&M. The logic for this proposition is evident in the assets constructed after 2011, where non-functionality without a service provider is considerably higher than those with service providers (see Figures 3 and 4, respectively). Baumann [43] describes how hand pumps only last around 5 years without appropriate O&M within traditional CBM approaches, and therefore the premature non-functionality of these assets within this timeframe is potentially attributed to the quality of the initial borehole construction (e.g., the high rates of partial functionality and non-functionality in the Figure 4 data for 2011–2016). As there are no service providers present to conduct appropriate O&M or collect funds for appropriate O&M, attention falls on the quality of the infrastructure in place of these newly constructed assets. In particular, 2012 and 2013 saw a significant increase of installations (see Figure 2) and revealed a particularly high non-functionality rate within this 5-year timeframe.

Annual or seasonal variations in groundwater levels have also been found to contribute to a decreased level of service provision across Africa [8,21,37,45]. Boreholes constructed during the wet season, high groundwater level conditions, are often constructed shallower and can go dry during dry season, low groundwater level conditions. However, groundwater level variations contributing to water point downtime should not be considered an acceptable risk when implementing water supply infrastructure, but rather poor hydrogeological oversight during planning and construction [10]. This further suggests that rapid provision of water supply has impacted the sustainability of water supply infrastructure, contradicting national policy requirements for sustainable coverage at the rural level [28]. Successes from other African countries found installations drilled to combat seasonal changes in the dry season were more reliable [45], as deeper water points increased sustainability and climate resilience [14].

The Afridev hand pump boreholes constructed after 2011 (within the 5-year margin previously described by Baumann [43]) without service providers to provide the appropriate O&M measures indicate potential sub-standard supplies. The data in Figure 3 of late MDG era supplies further endorse this. After 2011, the functionality distribution remained relatively consistent between 2012 and 2016, when it was assumed that well-constructed assets would express a higher rate of functionality than newer assets. This suggests that poor quality borehole installation and seasonal water levels are

potentially causing functionality issues rather than hand pump O&M, a relationship expressed across other African countries [8]. While issues such as poor O&M or willingness to pay factors cannot be ruled out at this stage, the partial functionality and non-functionally of these early assets are potentially problematic. This is notable in 2015, at the end of the MDG era, when very new assets expressed high rates of partial and non-functionality (14.52 and 16.13%, respectively) compared to adjacent years (see Figure 4).

This theme is supported by the findings of Mannix et al. [10] in their related CJF forensics examination of boreholes, where water source and borehole issues in Malawi were seen to contribute strongly to a reduced service provision. Notably, the main findings highlighted that many functionality problems were a symptom of water resource issues (72% of all cases) and borehole and installation issues (72% of all cases). There were fewer cases where hand pump parts were the cause of impacted performance (24% of all cases), highlighting poor O&M through decentralised management. Foster et al. [37] further describe this issue in regards to hydrogeological impacts on functionality. Both studies indicate that hydrogeology and borehole installation quality are a potentially permanent root cause of reduced functionality and to reduced service, even with routine O&M for water supply assets, which further highlights the need for improved standard of work prior to installation and during construction of the borehole.

Infrastructure sustainability suffers when technical oversight of borehole construction is ignored by donors and NGOs [7,10,46]. Implementing organisations have a responsibility to follow national standards and ensure systems are fit for their purpose, as implementing sub-standard boreholes undermines the policies of the Government of Malawi, and ultimately SDG 6. Problems may similarly arise from inappropriate commissioning of boreholes if water supply quality concerns, notably salinity in parts of alluvial aquifer systems in southern Malawi [47], are ignored or overlooked while prioritising coverage.

The "business as usual" investment into infrastructure that has prioritised the MDG coverage targets over quality of infrastructure is an issue entering the SDG era, especially where it has become evident of premature breakdown within a few years of their installation. Furthermore, the rehabilitation exercises conducted on these assets installed during this timeframe highlight potentially sub-standard infrastructure not exclusively limited to the hand pump (especially where rehabilitation was carried out resulting in partially functional or non-functional supplies—see Figures 3 and 4). As previously mentioned in Section 2.3, rehabilitation is considered the start of a new service.

Our findings, alongside wider evidence cited, suggest there is an underlying issue of infrastructure installation quality that has an impact on functionality and sustainability. While this is most notable of the Afridev hand pump boreholes installed after 2011, there is a risk that this issue has been present across the whole MDG period. These problems pose a risk for Malawi in the transition between the MDGs and SDGs, and while the re-evaluation of O&M dominates discussions, it is imperative that the quality of borehole construction also be improved. The drive to continually provide new assets and implement community-focused hand pump boreholes across Malawi for coverage targets must be reconsidered within monitoring the success of the SDG agenda. This is further complicated by the challenges of CBM and the lack of capacity to contend with the challenges of O&M and CapManEx. Proactive approaches to O&M and CapManEx reduce the risk of breakdowns that require costly rehabilitation to maintain operational levels necessary to the life cycle of the system [23]. However, with the challenges of conducting the necessary CapManEx, rehabilitation is more likely to be needed, and less likely to be conducted within CBM.

*3.4. Rehabilitation Conducted on MDG Assets*

Out of the 844 recorded cases of Afridev hand pump borehole rehabilitation (see Figure 5), much of the work was conducted close to the end of the MDG era into the start of the SDGs between 2014 and 2017 (13.74, 15.05, 22.63 and 24.17% of MDG constructed hand pump boreholes, respectively). Between 2000 and 2013 less than 5% of recorded rehabilitation cases were conducted each year.

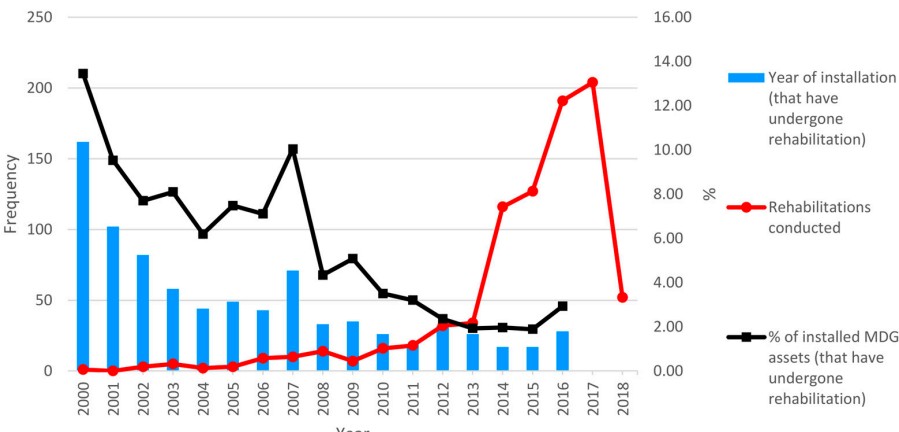

**Figure 5.** Number of rehabilitation exercises conducted on MDG Afridev hand pump boreholes (n = 844).

This data set showing a progressively greater number of rehabilitation exercises targeting early MDG installations (normalised percentage curve, noting the unexplained 2007 peak) further supports the impact depreciation has on the functionality of these assets. This is to be expected, as Afridev hand pumps installed in the boreholes that received regular maintenance from a service provider had typically reached the end of their expected design life. Where no service provider was present (see Figure 4), depreciation was more visible. When service delivery ceased over a number of years, rehabilitation should be considered as the start of a new service and a new capital expenditure, an often-neglected cost in pre- and post-construction of rural water supply infrastructure [7,48].

However, the investments that increased the coverage of the MDGs risked masking an underlying issue with further implications concerning the SDGs. When considering life cycle costs, the impacts of sub-standard borehole infrastructure must be considered alongside routine hand pump maintenance. This further supports the need for professional hydrogeological oversight during construction, and proactive approaches in both routine O&M and CapManEx to maintain adequate service delivery and prevent or prolong failure from further depreciation or abandonment of infrastructure (particularly where major repairs through CapManEx are lacking, as indicated by the partial functionality in Figure 3). The risk of waiting until major repairs are needed results in rehabilitation, which further leads to reliance of external support to remediate the service delivery.

### 3.5. Reliance on External Support

Malawian policy recommends borehole construction and rehabilitation of water supply systems to the international aid and private sectors [29]; however, these organisations often have their own agendas when considering service delivery [49]. Further pressures to achieve more sustainable solutions that are less reliant on external support will prove necessary with any decline of international funding (e.g., the substantial reduction of the UK's Department for International Development planned budget for overseas aid in Malawi between 2017 and 2020 [50,51]).

The "one-time investment" for donor-funded rural water supplies requires practitioners to follow proper installation standards but only covers the immediate need of rural communities, as it does not provide prospects for sustainability or growth [8,33,49]. Once external support is withdrawn, local government is relied upon to provide support which can be beyond their capacity to maintain the level of support required. Table 2 presents the funding actors for the rehabilitation exercises previously explored (see Figures 3 and 4). Rehabilitation exercises have primarily been provided by NGOs accounting for over 504 (59.72%) of all cases, clearly supporting that reliance on external support for rehabilitation is present in rural Malawi.

**Table 2.** Frequency of actors funding rehabilitation.

| Funder | No. of Rehabilitations Conducted | % of Total Rehabilitations Conducted |
|---|---|---|
| Community | 145 | 17.18 |
| Local Government | 54 | 6.40 |
| NGO | 504 | 59.72 |
| Politician | 52 | 6.16 |
| Religious Institution | 20 | 2.37 |
| School | 12 | 1.42 |
| Water Point Committee (WPC) | 22 | 2.61 |
| Other | 32 | 3.79 |
| Don't know | 3 | 0.36 |
| Total | 844 | 100 |

In the previously mentioned cases of sub-standard construction and lack of proactive approaches, repeat rehabilitation is often required to bring a supply back up to operational condition, which is an enormous waste of investment [7] and an unsustainable solution to depend on for the sector.

A study by Chowns [36] found that CBM worked as a method, disseminating responsibility from the government and funders to the community, but in practice had failed to deliver any technical or financial benefits. CBM relies on local governments that have limited resources and do not establish efficient support for their communities [43,52,53], which often leaves decentralised service providers dependent on external institutions for long-term financial sustainability, such as the private sector or NGOs. While a community may be able to finance and implement minor repairs, major repairs may present more of a challenge due to the substantial costs involved for rural decentralised service delivery. Many communities must rely on external support such as NGOs to provide necessary financial support to bring the hand pump boreholes back to an operational state. A similar situation is found in Zimbabwe, where NGOs are the sole funders of rehabilitation [15]. However, in Malawi, the local governments are expected to provide the support to service providers once external support has left after implementation. With Malawi's own National Water Policy promoting CBM alongside consultation with local governments, necessary and costly rehabilitation exercises are problematic. Notably in Table 2, communities have funded rehabilitation in more instances than local governments, meaning local economies and initiatives can assist with the costs of maintenance [54]. This further highlights the financial challenges local governments face in Malawi's rural sector when considering post-construction support, also emphasised by Baumann [43].

While NGOs have been the primary actors in funding rehabilitation, there are considerable risks associated with relying on external support. It has been previously established in literature that donors and NGOs may ignore or undermine national policies in favour of project-orientated results, resulting in a disregard for government-led priorities for the long-term sustainability of water supply and capacity building/institutional support. Furthermore, rehabilitation that has been conducted on assets at the beginning of the MDGs (see Figure 5) where no service provider was present should be considered a poor investment that sacrifices sustainability for coverage. These findings have implications for low-income regions moving into the SDG era, particularly where national policies reflect coverage [14] and decentralisation of the rural water sector [15,33,55].

There is no quick fix for rural water supply sustainability and long-term investment planning based on sustainability is required [7,34,49]. Despite the wide international push from stakeholders to increase coverage in the sector, professionals and practitioners have contributed to the problem of failing water supplies [7] and decentralisation, most notably CBM, which has contributed to the burden on the rural water supply sector.

## 4. Conclusions

Over the duration of the MDG era, there has been a positive shift in the coverage of rural water supplies. However, the implementation of water supplies has been subject to the influence of national policies and the MDGs that have induced some acceleration of activity to meet coverage targets. The evidence presented indicates that the acceleration towards meeting coverage targets contributes to sustainability challenges within the MDGs, with further implications moving into the SDGs. This provides grounds for water policy guidance updates on minimising the impacts to long-term sustainability.

The drive for decentralisation or "community-led" management of these rural water supplies has left the rural populations of Malawi with the burden of maintaining these assets. However, it is well established that service providers struggle to provide the maintenance and major repairs required to keep services operational sustainably. The reactive approach to the O&M and CapManEx of supplies contributes to the decline of functional assets, which is compounded by the notable effect of depreciating infrastructure across the MDG era. This has produced a growing need for rehabilitation exercises to bring the supplies that were implemented primarily during the early MDG era back to an operational standard. Proactive approaches to adequately maintain these supplies are necessary to prevent or postpone these costly rehabilitation exercises, which are an unsustainable practice due to their reliance on external support and the limited capacity for local governments to fund.

Furthermore, the investments into meeting the coverage targets of the MDGs, in particular those installed after 2011, suggest underlying issues that are not exclusively a product of the hand pump. Non-functionality across newly constructed assets, and some rehabilitation of newly constructed assets, points to a risk of sub-standard quality of infrastructure that could potentially impact assets installed across the MDG timeframe. These risks further contribute to the burden of maintaining assets that are inherently unsustainable at a decentralised level, and will have a further impact upon the progress of the SDGs.

This has implications for long-term sustainability transitioning in the SDG 2030 agenda due to the limited communication between NGOs and local governments/service providers, leading to a lack of reporting of reasons for partial functioning or non-functioning services, and ultimately in poorly targeted investments. The resulting decline in service delivery has the potential risk of an increased usage of unimproved supplies that could lead to issues hidden by the MDG coverage targets, such as impacts to the health of the rural population or an increased user burden for a neighbouring improved supply.

These lessons learned from Malawi have significance for other low-income regions, particularly those in SSA that rushed to meet or fell behind MDG 7c by 2015. National policies that focus on coverage and decentralisation of rural water supplies will potentially be subjected to the same challenges as Malawi of sustainably maintaining infrastructure. Moving towards 2030, lessons must be learnt from the evidence in Malawi of the coverage target approach in the MDGs to ensure that low-income regions are not further subjected to the similar risks of unsustainable water supply infrastructure. The evidence points to a burden of maintaining aging assets sustainably, combined with the several occurrences of sub-standard boreholes that have added to the established complexity of managing rural water supplies, as the implementation of assets to meet coverage targets points to justified concerns of sustainability.

It is recommended that conducting the frequent monitoring of water supply assets is essential to the success of the SDGs, and for proactive post-construction support for service providers to achieve sustainable investments. The impact from the notable peak of installations in the late MDGs is not fully evident yet, as the elapsed time is five years. Further research into ongoing monitoring of these assets is required to establish the full impact on communities entering the 2030 agenda, and to improve the performance of service provision at the local level rather than solely measuring it. Further research into proactive approaches at the local level that support the capacity for building is required to address

the burden of water supply infrastructure in low-income regions. This is ongoing and will be the subject of subsequent papers.

**Author Contributions:** Conceptualization, J.P.T.; Methodology, J.P.T. and A.V.M.M.; Formal Analysis, J.P.T.; Data Curation, J.P.T., A.V.M.M., M.N., P.M. and R.M.K.; Writing—original draft preparation, J.P.T.; Writing—review and editing, J.P.T., A.V.M.M., N.M., M.N., M.O.R., A.B.C. and R.M.K.; Visualization, J.P.T. and A.V.M.M.; Supervision, A.B.C. and R.M.K.

**Funding:** This research was funded by the Scottish Government Climate Justice Fund Water Futures Programme research grant HN-CJF-03 awarded to the University of Strathclyde (R.M. Kalin).

**Acknowledgments:** The authors would like to acknowledge support from the Scottish Government Climate Justice Fund Water Futures Programme, and the University of Strathclyde. We thank continued direct collaboration with our partner the Government of Malawi Ministry of Agriculture, Irrigation and Water Development (MoAIWD), and implementation partners United Purpose, World Vision, CADECOM, CARE, BAWI and Water for People for their support. In particular, the authors would like to thank BASEflow for their ongoing technical and logistical support.

**Conflicts of Interest:** The authors declare no conflict of interest.

**Appendix A**

As described in Section 2.3, the 14,943 drilled boreholes equipped with Afridev hand pumps installed between 2000–2016 were captured from the 68,984 water points mapped by the CJF to date (January 2019), which are expressed in Figure 1. Table A1 describes the districts and regions in Malawi where these Afridev hand pump boreholes were located, and the breakdown that made up the total subset.

**Table A1.** Locations and distribution of Afridev hand pump boreholes installed between 2000–2016 mapped to date by the CJF using the MIS platform mWater (as of January 2019).

| District of Malawi | Region of Malawi | n | % of Total Data Set |
|---|---|---|---|
| Balaka | Southern | 817 | 5.47 |
| Blantyre | Southern | 1048 | 7.01 |
| Chikwawa | Southern | 967 | 6.47 |
| Chiradzulu | Southern | 530 | 3.55 |
| Dedza | Central | 1130 | 7.56 |
| Dowa | Central | 392 | 2.62 |
| Karonga | Northern | 1 | 0.01 |
| Kasungu | Central | 515 | 3.45 |
| Likoma | Northern | 3 | 0.02 |
| Lilongwe | Central | 2308 | 15.45 |
| Machinga | Southern | 545 | 3.65 |
| Mangochi | Southern | 1724 | 11.54 |
| Mchinji | Central | 29 | 0.19 |
| Mulanje | Southern | 437 | 2.92 |
| Mwanza | Southern | 335 | 2.24 |
| Mzimba | Northern | 17 | 0.11 |
| Neno | Southern | 8 | 0.05 |
| Nkhotakota | Central | 327 | 2.19 |
| Nsanje | Southern | 347 | 2.32 |
| Ntcheu | Central | 808 | 5.41 |
| Ntchisi | Central | 65 | 0.43 |
| Phalombe | Southern | 514 | 3.44 |
| Salima | Central | 177 | 1.18 |
| Thyolo | Southern | 850 | 5.69 |
| Zomba | Southern | 1004 | 6.72 |
| Border Community/Outside Malawi [1] | | 45 | 0.30 |
| Total | | 14943 | 100% |

[1] These Afridev hand pump boreholes were located across, or on the border of Malawi. These were assets owned by the Government of Malawi and managed by a Malawian Community.

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
