# Peer review of "Understanding the Functionality and Burden on Decentralised Rural Water Supply: Influence of Millennium Development Goal 7c Coverage Targets"

_water, doi:10.3390/w11030494_

Round 1

Reviewer 1 Report

The manuscript provides interesting analysis and has the potential to be an important contribution towards efforts to improve the sustainability of rural water services in Africa. 

In general, I think the article would benefit from (a) clearer presentation of the data in Table 1 and Figures 3 & 4, and (b) a clearer evidential or logical bases for linking the results with quality of construction and seasonality (as opposed to poor O&M or collective action or willingness to pay issues)

Specific comments:

Line 119: “However, the monitoring of the data suggests a greater than 95% confidence rate.” – Would be useful to know exactly what was monitored and how (e.g. revisiting a sub-sample of water points? Checking data as it was uploaded to server?) And how did you come to conclusion of 95% confidence rate – is this in the literal sense (e.g. >95% of data points that were double-checked were accurate?).

Line 135: “Therefore, for avoidance of doubt, the term ‘Functional’ is used here to describe a water point 136 that is in good condition and providing water in line with the design specifications” – this is still a vague definition – how was “good condition” determined? Were there clear criteria for assessing the condition, or was it a subjective assessment by enumerators?  Equally, what are the design specifications for the Afridev handpump against which the functionality was assessed? (e.g. was it based on the Afridev’s discharge vs depth curve?)

Line 148: “This resulted in a dataset of 14,943 Afridev handpump 149 boreholes” – Just to be clear – are these all actually drilled boreholes? Does this include or exclude hand-dug wells that are equipped with Afridevs? 

Line 164: “Within this dataset, rehabilitation is defined as a single major repair consisting of 1,500,000MK 165 (Approximately 2062USD)” – how was this threshold decided? Seems quite a high benchmark as in most countries this is about double the cost of a new Afridev handpump with its max pumping lift. So does this mean you only consider an activity to be a rehab if it also rehabilitates the borehole itself?

Line 171-173: “According to the JMP, Malawi has shown a positive shift in the rural water supply coverage with 85% coverage of 173 improved supplies (63% at least basic and 20% limited) between 2000 and 2015”. Sentence is a bit confusing. Was 85% the figure by end of 2015? And if so, what was the size of the increase between 2000 and 2015?

Table 1 is very confusing in its layout.  First, rather than super-script numbers to delineate rows (i.e. with/without rehab), can you just use the labels in the row themselves. Secondly, I understand functionality rate of 71% (service provider present).  But I don’t understand how you can get a median, min and max functionality rate given functionality is a binary variable.  Equally, the table seems to suggest that with rehabilitation conducted and service provider present, the functionality rate is 4.86% which does not seem plausible.

Figure 3 – It took me quite a while to interpret this chart with the dual y-axes (especially because the functionality bars don’t add up to 100% - I think I worked out why, but it took a while because the NF, PF, F are not expressly specified as ‘non-rehabilitation’ whereas maybe the should be??). For ease of understanding, I would suggest trying to re-configure/redesign this chart to make them more intuitive for the reader. Same applies to Figure 4.

Line 347: “import” should perhaps be “important”?

Line 351: I don’t quite follow how the non-functionality rate from 2015 is specific evidence of poor quality borehole installation and seasonality (as opposed to O&M). Couldn’t it also be that a certain proportion of communities are unwilling or unable to collect money and pay for the repair when the system broke down? I think you need more empirical evidence to make this link or at least provide clearer logic to justify your conclusion that the data presented specifically demonstrates poor quality construction and seasonality.

Author Response

Thank you very much for the expert comments and suggestions in the review of this manuscript. We have responded positively to all the comments and believe the suggestions put forward have greatly improved the content and overall clarity of the manuscript. 

To this end we have reconfigured the presentation of the suggested Table 1 and Figure 3 & 4 to clearly present the results as suggested “must be improved”, including improving the logic of the argument pointing towards quality of infrastructure. We have also improved the conclusions so that they are supported by the results in the discussion and conclusion section, as suggested “must be improved”. This also includes clearer recommendations for further research. 
Please see the pdf attached for specific comments. Changes to the manuscript have been highlighted using the “Tracked Changes” function in MS Word. Spelling and grammar checks have been applied to the manuscript as suggested “English language and style are fine/minor spell check required”.

Reviewer 2 Report

This paper investigates the accomplishment of SDGs through the balance in implementing Afridev handpump boreholes in Malawi. The topic is in line with the aims of the journal. Some suggestions should be taken into account in order to improve the readeness and originality of the paper.

A lot of references are cited more than once through the text. It would be more interesting to cite each reference only once and add some new references. 

Methodology is well described. The title of section 2.1 should be changed to "Collection data" in order to make reference to the different sources used in the analysis.

A brief section about the case study (Malawi access to water and sopciodemographic profile is needed in order to understand the starting point situation of the analysis: which is the gains and loses of the application of the SDG 6?

The main weakness of the paper is the poor quality of the content. Figure 1 (map) does not provide relevant information about the location of the handpump boreholds... A location map should be provided but not in this present form. Details about the location of the handpumps by region could be more interesting if you put this information in a table. This information should be in absolute number and in percentage, for example.

Figure 2 could be also improved. A bar chart could be more useful to represent data, also in absolute and % values. Figure 6 does not provide sufficient information to be represented in a figure, maybe the results could be added to the text.

A Discussion section must be provided in order to compare the study case of Malawi with other African countries, for example, in order to debate about the added-value of the policies and actions conducted to address the SDG 6 gap.

The Conclusions are well exposed and include some recommendations for future research.

Author Response

Thank you very much for the expert comments and suggestions in the review of this manuscript. We have responded positively to all the comments and believe the suggestions put forward have greatly improved the content and overall clarity of the manuscript. 

To this end we have reconfigured the presentation of the suggested Figures including transposing the data in Figure 6 into a Table as suggested. We have also improved the logic of the argument pointing towards quality of infrastructure, with references to other African countries. We have also improved the conclusions so that they are supported by the results in the discussion and conclusion section, as suggested “must be improved”.

Please attached pdf for specific comments. Changes to the manuscript have been highlighted using the “Tracked Changes” function in MS Word. Spelling and grammar checks have been applied to the manuscript as suggested “English language and style are fine/minor spell check required”.

Round 2

Reviewer 1 Report

Thanks for the thorough revisions. The paper is an important and thought-provoking contribution to the rural water sector. 

Reviewer 2 Report

Congratulations for achieving the suggestions expressed in the review report, a very good job.